# Germline JAK2 E846D Substitution as the Cause of Erythrocytosis?

**DOI:** 10.3390/genes14051066

**Published:** 2023-05-11

**Authors:** Nada Maaziz, Céline Garrec, Fabrice Airaud, Victor Bobée, Nathalie Contentin, Emilie Cayssials, Antoine Rimbert, Bernard Aral, Stéphane Bézieau, Betty Gardie, François Girodon

**Affiliations:** 1Laboratoire de Génétique Chromosomique et Moléculaire, Pôle Biologie, CHU de Dijon, 21000 Dijon, France; 2Service de Génétique Médicale, CHU de Nantes, 44000 Nantes, France; 3Service d’Hématologie Biologique, CHU de Rouen, 76000 Rouen, France; 4Service d’Hématologie, Centre Henri Becquerel, 76000 Rouen, France; 5Service d’Oncologie Hématologique, CHU de Poitiers, 86000 Poitiers, France; 6l’Institut du Thorax, INSERM, Nantes Université, CHU Nantes, 44300 Nantes, France; 7Ecole Pratique des Hautes Etudes, Université PSL, 75006 Paris, France; 8Laboratory of Excellence GR-Ex, Imagine Institute, 75015 Paris, France; 9Service d’Hématologie Biologique, Pôle Biologie, CHU de Dijon, 21000 Dijon, France; 10Inserm U1231, Université de Bourgogne, 21000 Dijon, France

**Keywords:** erythrocytosis, *JAK2* mutation, germline mutation, familial erythrocytosis

## Abstract

The discovery in 2005 of the *JAK2* V617F gain-of-function mutation in myeloproliferative neoplasms and more particularly in polycythemia vera has deeply changed the diagnostic and therapeutic approaches to polycythemia. More recently, the use of NGS in routine practice has revealed a large number of variants, although it is not always possible to classify them as pathogenic. This is notably the case for the *JAK2* E846D variant for which for which questions remain unanswered. In a large French national cohort of 650 patients with well-characterized erythrocytosis, an isolated germline heterozygous *JAK2* E846D substitution was observed in only two cases. For one of the patients, a family study could be performed, without segregation of the variant with the erythrocytosis phenotype. On the other hand, based on the large UK Biobank resource cohort including more than half a million UK participants, the *JAK2* E846D variant was found in 760 individuals, associated with a moderate increase in hemoglobin and hematocrit values, but with no significant difference to the mean values of the rest of the studied population. Altogether, our data as well as UK Biobank cohort analyses suggest that the occurrence of an absolute polycythemia cannot be attributed to the sole demonstration of an isolated *JAK2* E846D variant. However, it must be accompanied by other stimuli or favoring factors in order to generate absolute erythrocytosis.

## 1. Introduction

The term erythrocytosis is frequently used when increased values of hematocrit (Ht) and hemoglobin (Hb) are observed in patients in clinical practice. However, absolute erythrocytosis can only be defined by an increase in red cell mass above (RCM) 125% of the theoretical one [1]. Although RCM measurement is the most accurate definition of erythrocytosis (also called “polycythemia”), it is less and less available in many countries. In routine practice, hematological parameters (hemoglobin, hematocrit) are used as surrogate markers in the diagnosis of polycythemia [2].

However, these parameters do not depend exclusively on RCM. Indeed, changes in the plasma volume can alter hemoglobin and hematocrit levels even though the erythrocyte mass remains unchanged. These red cell diseases can be the result of multiple genetic causes; the main one being the acquired V617F mutation in the *JAK2* gene [present in more than 95% of Polycythemia Vera (PV) cases] causing constitutive activation of the JAK2 kinase in cell lines.

JAK2, a member of the Janus kinase family, has a structure formed with seven Janus homology domains (JH1-JH7). The activation of JAK2 is the result of its binding to hematopoietic cytokine receptors such as the erythropoietin receptor (EPOR), thrombopoietin receptor (MPL), or the granulocyte-macrophage colony-stimulating factor (GM-CSFR) and IL-3 and IL-5 receptors [3]. The activation of JAK2 in myeloproliferative neoplasms (MPNs) leads to the phosphorylation of the receptors at the cytoplasmic tail, thus creating docking sites for SH2 domain-containing signaling proteins such as the signal transducer and activator of transcription (STAT3 and STAT5) [4,5]. These phosphorylated proteins dimerize and translocate to the nucleus to regulate transcriptional programs leading to cell proliferation, differentiation, and survival. This activation is also the promotor of another pathway: the mitogen-activated protein kinase (MAPK), which is activated in other cancers [6]. This pathway is mediated by major protein kinases including RAF, MEK1/2, and ERK1/2, strengthening cell proliferation and survival. Regulators of the JAK2 pathway have been thoroughly described: suppressors of cytokine signaling (SOCS) proteins inhibiting JAK2 [7], phosphatases (SHP1 and SHP2) as well as other protein tyrosine phosphatases (PTPs) leading to JAK2 inactivation via dephosphorylation [8], LNK (also known as SH2B3) inhibiting erythropoiesis and thrombopoiesis [9].

The occurrence of somatic *JAK2* mutations has been very thoroughly described in MPNs, including PV, Essential Thrombocythemia (ET), and primary myelofibrosis [10]. These are heterogeneous diseases characterized by the proliferation of blood cells by hematopoietic precursors, leading to the disruption of hematological and histological homeostasis [11].

On the other hand, congenital erythrocytosis related to germline mutations, is rare, and usually diagnosed after extensive testing, including molecular genetic testing. When confronted to a new case of erythrocytosis, several aspects must be explored in order to clarify the diagnosis (Figure 1) [12]. Apparent reasons of erythrocytosis must be excluded: chronic hypoxia (due to respiratory failure, chronic obstructive pulmonary dysfunctions, sleep apnea, congenital heart defects…), tumors (kidney, uterine fibroma…). As such, several easy and cost effective analysis must be conducted: blood electrolytes, proteins, serum EPO, arterial/venous blood gas, abdomino-pelvic ultrasound, hemolytic and iron status, bone marrow biopsy and progenitor culture, respiratory functional explorations, polysomnography. The compilation of clinical and biological evidence can specify the origin of the disease and help distinguish congenital erythrocytosis from PV. However, recent studies have shown that germline mutations could also be found in MPN cases, misleading the diagnosis [13,14,15].

The diagnosis of such diseases is essential since they may be associated with severe thromboembolic events, pulmonary hypertension and, occasionally, tumors. These events can be prevented by the implementation of the appropriate treatment. When confronted with idiopathic erythrocytosis, the current guidelines formulated by the British Society of Hematology for therapeutic management consider (i) venesection to reduce the Ht if symptoms are associated with high Ht, or if previous thrombotic events (either in personal or familial history), (ii) low dose aspirin if otherwise clinically indicated for primary or secondary prevention [16]. The search for novel therapies still represents a critical issue since venesections can be hardly tolerated by some patients, and difficult to set up in routine practice [17].

As previously stated, the routine use of next generation sequencing (NGS) methods and whole exome/genome sequencing has shown light on other types of mutations in the *JAK2* gene involved in the pathogenesis of erythrocytosis. Indeed, germline variants in the *JAK2* gene have been reported to be a cause of familial erythrocytosis, such as the heterozygous germline *JAK2* E846D substitution [18]. However, in the family studied, the co-segregation of the *JAK2* E846D variant with erythrocytosis was restricted to a nuclear family in which the non-affected sibling could not be tested. This partly incomplete co-segregation analysis makes it difficult to draw solid conclusions.

This mutation has been described partly as the genetic cause of familial erythrocytosis, by an activation of the JAK/STAT signaling pathway. Indeed, Kapralova and al [19] reported a case of erythrocytosis of a young boy harboring two *JAK2* germline mutations at the heterozygous state *JAK2* E846D and *JAK2* R1063H, inherited from his mother (carrier of the *JAK2* E846D) and his father (carrier of the *JAK2* R1063H). In vitro experiments showed a weak constitutive activation of the JAK/STAT signaling pathway via the erythropoietin receptor (EPOR) by E846D and R1063D, although much lower than the activation by *JAK2* V617F, with R1063 showing even weaker activation. This weak activation could not lead to the expression of the erythrocytosis phenotype by the sole presence each variation. However, it was postulated that both mutations expressed at the same time could indeed lead to a sufficient increase in JAK2 activity resulting in the pathological phenotype. This was confirmed in in vitro colony assay using mice bone marrow cells retrovirally transduced with murine JAK2 expression vectors: a significantly higher CFU-E and BFU-E was noted in E846D mutant transfected cells compared with JAK2 wt transfected cells.

In our routine exploration of well-characterized erythrocytosis cases with NGS using a dedicated panel, we have been confronted with germline mutations in the *JAK2* gene. Most identified variants were polymorphisms (ClinVar and GnomAD database) and were already tested by in vitro experiments such as the *JAK2* G571S that was classified as non-causal in the development of MPNs [20]. Panovska-Stavridis et al. reported a family harboring a germline *JAK2* G571S mutation in the heterozygous state, but only one out of the four members of the family investigated presented a phenotype of chronic thrombocytosis. The affected protagonist was presented with hematological parameters raising the hypothesis of ET, and further confirmed by histologic examination showing and increased number of megakaryocytes in the context of sustained thrombocytosis (platelet counts > 1500 × 10^9^ per liter over a period of 3 months). A molecular investigation was initiated, with the exploration of *JAK2, CALR*, and *MPL* mutations, and showed that the patient presented two variants: a germline C-T substitution at codon 571 in exon 13 of the *JAK2* gene (*JAK2*G571S) and a somatic *CALR* mutation (insertion p.K385fs*47). The germline *JAK2*G571S variation was also present in three family members: the father, the brother, and the daughter, who had normal hematological parameters showing no ET phenotype. Once again, the familial segregation of the variation was restricted to non-affected family members.

The germline *JAK2* E846D variants brought to light by our routine NGS caught our attention, but it was complicated to link the sole presence of such variants with the pathological phenotypes explored.

To get a clearer picture, we report here our experience with two patients harboring an isolated *JAK2* E846D variant identified in a large national database of 650 patients with well-characterized idiopathic absolute erythrocytosis. In a second step, the erythrocyte parameters of patients from the UK biobank with this variant were analyzed.

## 2. Materials and Methods

### 2.1. Patients

Since 2016, a large French national cohort of 650 patients with well-characterized erythrocytosis has been prospectively recorded. Erythrocytosis was confirmed through clinical and biological tests (blood count, blood electrolytes, serum EPO, arterial and venous blood gases, screening of *JAK2* mutations in exons 14 and 12), and further exams including pulmonary function tests, abdominal ultrasound, and RCM measurements. All samples were screened for variants in genes involved in erythrocytosis using a dedicated NGS panel [21].

RCM determination was realized in the department of nuclear medicine of the patient’s hospital. Patients were informed not to fast before the examination and to be hydrated as usual. Blood samples were collected from each patient and the labelling of red blood cells (RBCs) with a known activity of technetium 99 was realized. The 99m-labelled autologous RBCs were then reinjected intravenously into patients, who were asked to remain rested for at least 15 minutes before reinjection. Three blood samples were collected in heparinised tubes, 15, 30 and 60 minutes after reinjection. The tubes were centrifuged and sampled without delay after each sample. Activity counts were performed at the end of the test on plasma and whole blood samples. Venous Ht was measured by microcentrifugation on each of the samples. Patients were asked to remain seated 15 min after the measurement and the last collection as well.

The RCM was calculated by the formula RCM (mL) = I × Ht/A0 where I (cpm) is the activity injected into the patient, Ht is the corrected venous hematocrit, and A0 (cpm/mL) is the activity of one milliliter of whole blood at equilibrium time, i.e., the average of the values at 15, 30, and 45 min. The technetium elution rate [EL(%)] was calculated according to the formula EL(t) = PL(t) × [1 − Ht(t)]/A(t) where PL is the activity of 1 mL of plasma. The elution values were extrapolated to time 0 [EL0]. The RCM corrected for elution was calculated according to the formula RCMcor = I × Ht × (1 − EL0)/Aeq [22].

### 2.2. In Silico Analysis

The UK Biobank resource [application number 49823 [23]], which gathers genetic and health information from half a million UK participants, was used to analyze the hematological parameters (red blood cells count, hemoglobin, reticulocyte counts) of the carriers of *JAK2* E846D genetic variants. The data collected by UK Biobank was the result of whole-exome sequencing (WES) extracted from deep genetic and phenotypic data collected on 500,000 individuals from across the United Kingdom, aged between 40 and 69 years old. DNA was extracted from blood samples collected from ~10% of the UKB individuals (50,000) included in the initial cohort, and whole exome sequencing was realized. The identified variants were the result of the work of dedicated bioinformatic pipelines. Variant Allele Frequency (VAF) were detailed for each identified variant [24,25].

### 2.3. Next Generation Sequencing

All patients included in the study have signed written informed consents. Blood samples were collected for diagnostic purpose after exclusion of classical causes of erythrocytosis (PV or secondary erythrocytosis associated with particular renal, cardiac or pulmonary disorder). DNA was extracted and quantified using a Qubit 3.0 fluorometer (Life Technologies-Thermo Fisher Scientific, Saint Aubin, France).

Molecular screening was performed by high throughput sequencing with MiSeq FGx System (Illumina, San Diego, CA, USA) using a dedicated NGS panel of genes involved (i) in the regulation of the hypoxia pathway (PHD1 [*EGLN2*], PHD2 [*EGLN1*], PHD3 [*EGLN3*], HIF-1A, HIF-2A [*EPAS1*], HIF-3A, *VHL*, *VHLL*); (ii) in the proliferation and differentiation of erythroid progenitors (*EPO*, *EPOR*, *JAK2*, LNK [*SH2B3*], *CBL*); (iii) or in mature cell function (bisphosphoglyceratemutase ([*BPGM*]) [26]. Bio-informatic tools were used to select the variants of interest, and to annotate the generated files.

### 2.4. Sanger Sequencing for Familial Segregation Analysis

Blood samples of the proband’s children were collected for familial segregation purposes. All patients have signed written informed consents. DNA was extracted and quantified using a Qubit 3.0 fluorometer (Life Technologies-Thermo Fisher Scientific, Saint Aubin, France), and Sanger sequencing was performed using a Big Dye Terminator v3.1 Cycle Sequencing Mix (Thermo Fisher Scientific, Waltham, MA, USA) and run on an Analyzer ABI PRISM 3130 XL (Thermo Fisher Scientific). Electrophoregrams were visualized through the software SeqAnalysis^®^.

## 3. Results

### 3.1. Patients

Among the 650 tested patients in the cohort with NGS sequencing, an isolated germline heterozygous *JAK2* E846D substitution was observed in two cases (NM_004972.4:c.2538G>C leading to a missense variation NP_004963.1:p.(Glu846Asp). NGS sequencing was performed with a dedicated panel for erythrocytosis, including the entire *JAK2* gene, all genes involved in hypoxia pathway and major genes involved in erythropoiesis. No other variation associated with the erythrocytosis phenotype was identified by the analysis, that screened for germline variations (VAF = 50%) and somatic variations (VAF < 50%).

The first patient was a 65-year-old man with no particular medical or family history of erythrocytosis, followed for absolute erythrocytosis (RCM = 135%, Hb = 19.1g/dL, Ht = 56%) since 2009. No mutation in the exons 12 and 14 of the *JAK2* gene was identified. The patient presented normal erythropoietin levels before phlebotomy (EPO = 6.7 UI/L, normal range (2.6–9 UI/L)), no histological argument on bone marrow biopsy for a possible MPN, and presented a negative complete etiological workup (in particular, no argument for a respiratory disorder, sleep apnea syndrome or tumor). This patient smoked (one pack/day for 40 years) and has tried to temporarily stop smoking several times (SaO_2_ = 96%, HbCO < 1% at the time of analysis). He has been treated since 2009 with phlebotomies (between four and eight/year) in association with low-dose aspirin. No thrombosis was noted in his history, neither before nor after the beginning of his phlebotomy and aspirin treatments.

A familial segregation analysis was performed, seeking familial erythrocytosis. His three children were explored: twins (one daughter, one son) and another son (Table 1). Genetic sequencing exploring the *JAK2* E846D variant was performed by using Sanger sequencing. The pedigree (Figure 2A) showed no familial history of erythrocytosis.

Sanger analysis identified a single nucleotide substitution at the heterozygous state NM_004972.4:c.2538G>C leading to a missense variation NP_004963.1:p.(Glu846Asp) in two members of the family: the daughter D1 and one of the two sons S2 (Figure 2B). Both affected members were healthy; no history of thrombosis was ever reported. No increase in Hb or Ht was associated with the presence of the heterozygous *JAK2* E846D variation (Table 1).

The second patient was a 51-year-old man whose father had “too much blood”. Unfortunately, no exploration was conducted for the father who died at 61 years old from a stroke. The medical history of the patient included well-balanced high blood pressure, high cholesterol, and thrombosis complicated by erysipelas on a leg two years before the diagnosis of erythrocytosis. He stopped smoking one month before the fortuitous discovery of the erythrocytosis. According to the patient, the erythrocytosis had been present for several years but not explored. His body mass index was 23.9 kg/m^2^, and the RCM was slightly increased (+22%) but not sufficient to achieve the diagnosis of absolute erythrocytosis. No renal or pulmonary abnormalities were found. The functional respiratory tests showed normal lung volumes and flows without alteration of the diffusing capacity. There was no splenomegaly or sign of MPN on the bone marrow biopsy. The full-body CT scan was normal. The *JAK2* E846D substitution observed in the peripheral blood was also found at the heterozygous state in the buccal swab, confirming its germline nature. As for the first patient, the serum EPO was normal (4.4 UI/L, normal range (2.6–9 UI/L)). The complete etiological workup found no explanation for the erythrocytosis, and the only variant identified by NGS analysis was the heterozygous *JAK2* E846D substitution. Because of the cardiovascular risk factors associated with erythrocytosis, a treatment associating phlebotomies and a low dose of aspirin was initiated.

No spontaneous thrombotic events were reported for either patient.

### 3.2. In Silico Analysis

An in silico analysis of the identified *JAK2* E846D variant with the large-scale biomedial database UK biobank was performed (gathering genetic and health information from half a million UK participants). The *JAK2* E846D variant was found in 760 individuals (347 men and 413 women) almost exclusively in heterozygous state (only one homozygous woman). The curves of values obtained for the carriers of the variants are almost overlapping the curves of the wild type individuals. Compared to controls, slight significant increases in Hb (13.6 versus 13.5 g/dL, *p* = 0.034) and Ht (39.2 versus 39.2%, *p* = 0.052) were noted in women carrying the *JAK2* E846D variant, whereas the increases in Hb and Ht were not significant in men (15.1 vs. 15.0 g/dl and 43.5 vs. 43.3%, respectively, *p* > 0.05) (Figure 3A). Furthermore, this variant was significantly associated (*p* < 5 × 10^−8^) with higher red blood cell counts but not reticulocyte count or Hb (Figure 3B), suggesting a potential role of the variant in erythropoietic stimulation but to a lesser degree than observed with the driver *JAK2* V617F mutation.

## 4. Discussion

Uncertain results have been reported so far regarding the implication of the *JAK2* E846D variant in the pathogenesis of absolute erythrocytosis. The variant was first described in a study, using in vitro and in vivo experiments, as a contributing factor leading to the activation of the JAK/STAT pathway via EPOR solicitation. A young case of erythrocytosis harboring two germline *JAK2* mutations at the heterozygous state *JAK2* E846D and *JAK2* R1063H was described. In vitro experiments showed weak constitutive activation of the JAK/STAT signaling pathway by E846D and even weaker activation with the sole presence of R1063H, that could not activate sufficiently the signaling pathway to express the phenotype. This was confirmed in an in vitro colony assay using mice bone marrow cells retrovirally transduced with murine JAK2 expression vectors: a significantly higher CFU-E and BFU-E was noted in E846D mutant transfected cells compared with JAK2 wt transfected cells [19]. In a more recent article, two sisters were referred for assessment of erythrocytosis, which lead to a familial exploration including their mother. NGS analyses were made independently in separate laboratories and revealed the presence of the germline *JAK2* E846D variant in all three members of the family in the heterozygous state [18]. Once more, this variant was portrayed as a contributing factor leading to the explored phenotype since no clear evidence could directly link its presence with the occurrence of erythrocytosis. However, in the family studied, co-segregation of the *JAK2* E846D variant with erythrocytosis was limited to a nuclear family in which unaffected siblings could not be tested. This partially incomplete segregation analysis does not allow strong conclusions to be drawn.

Thus, we used our cohort of 650 patients with thoroughly explored erythrocytosis cases to explore the causality of the *JAK2* E846D variant. NGS using a dedicated panel of erythrocytosis revealed 2 patients with heterozygous *JAK2* E846D variants with a VAF = 50%. For patient one, a familial segregation was realized in affected and non-affected siblings, whereas patient two had no family members to explore. This familial segregation of the variant carried by the family members showed the presence of the variation in two patients with normal range values of hematocrit and hemoglobin, suggesting no impact of the sole presence of *JAK2* E846D variant on erythropoiesis.

Low serum EPO have been reported in 2/3 of PV patients [27]. In a routine practice, decreased values of serum erythropoietin EPO points to the primitive character of erythrocytosis, suggesting the role of driver *JAK2* mutations in the occurrence of erythrocytosis. The same postulate has been proven in cases of erythrocytosis related to *EPOR* mutations, showing values of serum EPO often decreased [9]. In our study, the two patients with the *JAK2* E846D variant had normal serum EPO values, which argues against the implication of this variant in the presented phenotype, although it cannot be formally excluded either. Similarly, PVs with *JAK2*-atypical mutational profiles (such as the *JAK2* L611V mutation) [28], have decreased EPO values [29], as do PVs mutated for *JAK2* exon 12 [30]. The presence of normal EPO values in our 2 patients does not argue for a major impact of the *JAK2* E846D mutation alone in the occurrence of absolute erythrocytosis. Finally, additional data were collected from the UK Biobank analysis, based on the analysis of WES of a population of 50,000 individuals in the UK population. These data showed no elevation in Ht and Hb values in heterozygous carriers of the *JAK2* E846D variation, confirming the postulate that the sole presence of the variation could not cause absolute erythrocytosis.

A similar analysis was performed with the use of NGS with a targeted panel for erythrocytosis performed on patients with idiopathic erythrocytosis (IE) negative for canonical *JAK2* mutations and for secondary causes. It revealed a frequent association between the *JAK2* GGCC haplotype, a germline combination of single nucleotide polymorphisms (SNPs) and a *CALR* SNP (rs1049481, G > T) with an unknown biological significance in cases of erythrocytosis [31]. This was further confirmed by data resulting from an in silico variant from the 1000 Genomes Project (1000G), showing that the G allele (rs1049481_G) was significantly more frequent in patients than in controls, and in those harboring the *JAK2* GGCC haplotype. Furthermore, the genetic status of these patients harboring the *JAK2* GGCC haplotype was associated with additional somatic mutations affecting epigenetic regulators and transcription factors identified using NGS with a customized panel including 26 myeloid genes. Thus, this study showed once more the possible implication of the presence of SNPs and haplotypes, as contributing factors in the development of erythrocytosis and MPNs.

In a later study, a larger cohort of 80 patients with IE was used to further investigate the molecular complexity related to the occurrence of IE. Clinical and genomic data were explored to try to build the typical portrait of patients with IE, to facilitate routine explorations [32]. It was suggested that some cofactors created a predisposition for the development of erythrocytosis, and a hierarchy needed to be respected in the exploration process.

Altogether, our data from our routine NGS analysis on a large national database of 650 patients and the data collected from the UK Biobank database confirm by a complementary approach the results of Kapralova et al concerning the involvement of the *JAK2* E846D substitution in the genesis of erythrocytosis. As a consequence, the occurrence of an absolute polycythemia cannot be attributed to the sole demonstration of an isolated *JAK2* E846D variant. However, it must be accompanied by other stimuli or favoring factors in order to generate absolute erythrocytosis.

## Figures and Tables

**Figure 1 genes-14-01066-f001:**
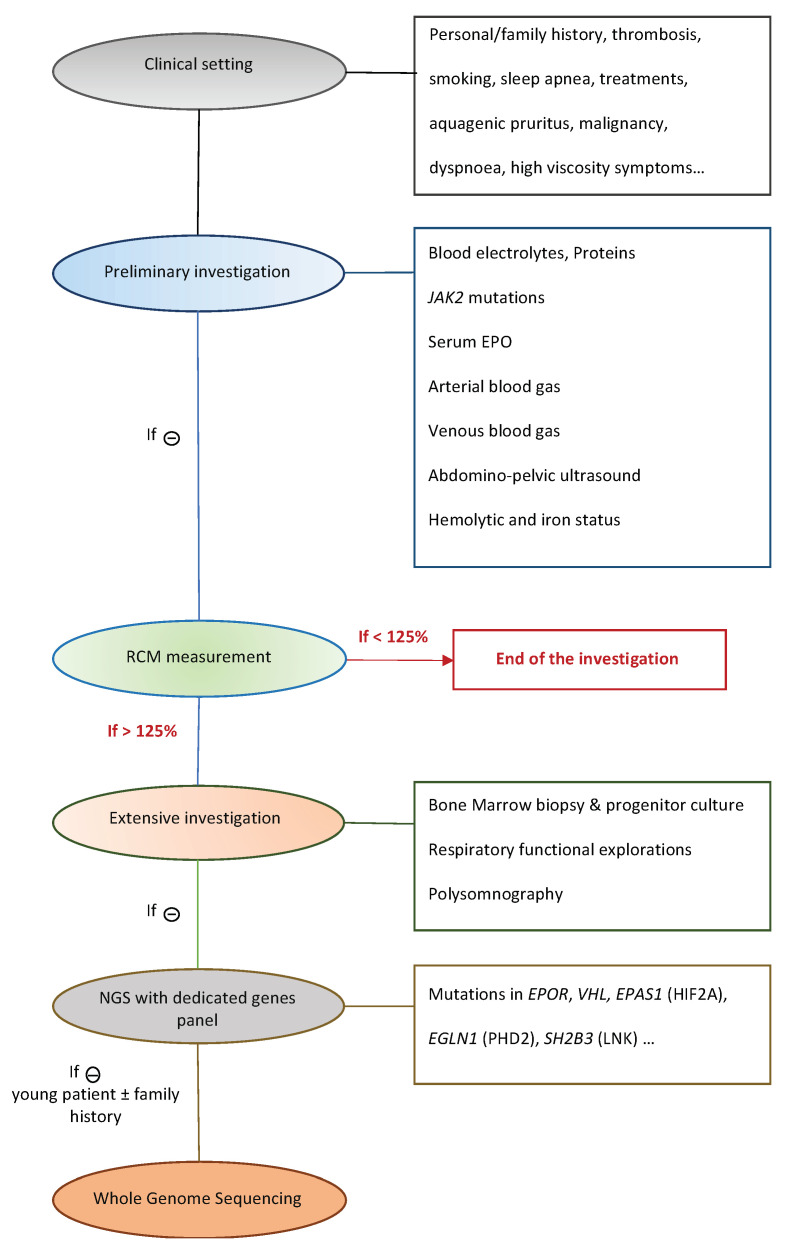
Diagnostic approach to erythrocytosis cases in Dijon’s hospital.

**Figure 2 genes-14-01066-f002:**
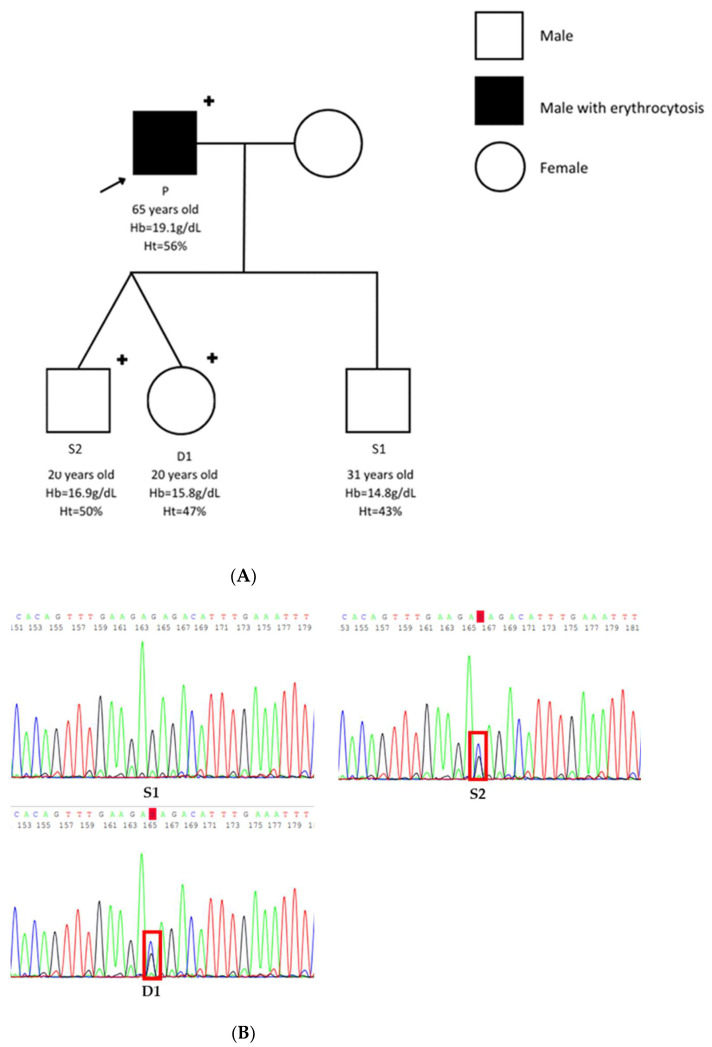
Presentation of patient one’s family. (**A**) Pedigree of patient one. (**B**) Electropherogram of the wild type son (S1), and mutated son (S2), mutated daughter (D1).

**Figure 3 genes-14-01066-f003:**
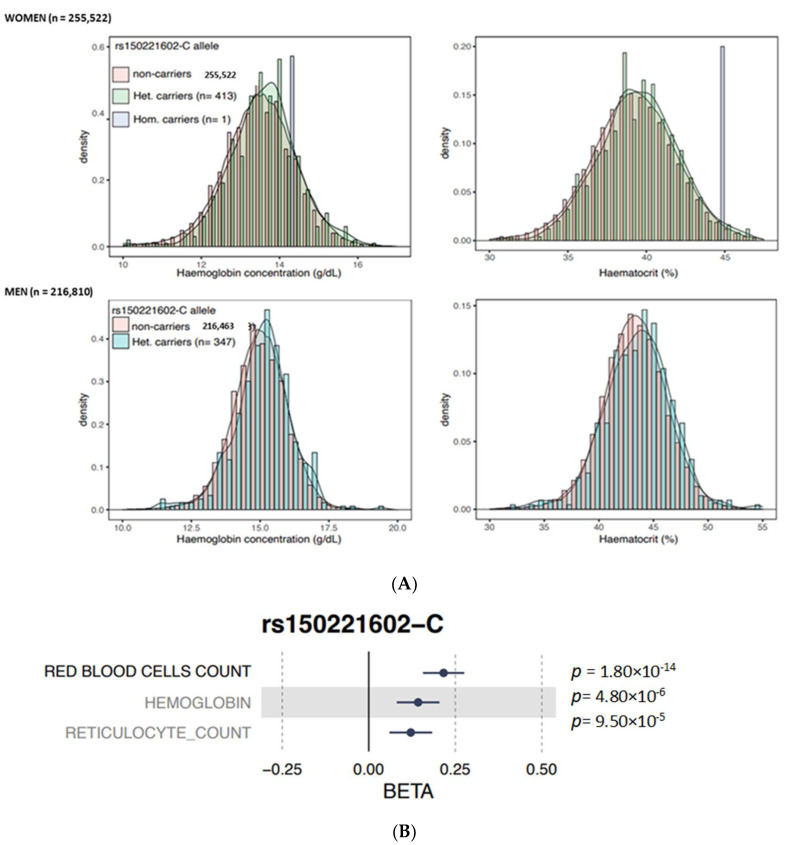
Illustration of the impact of the *JAK2* E846D variant on hematological features in men and women in the UK biobank database. (**A**) The colored bars represent non carriers of the *JAK2* E846D variant (pink bars) and the carriers of the mutation (green, and blue bars). (**B**) Impact of the *JAK2* E846D variant on hemoglobin, red blood cell count, and reticulocyte count.

**Table 1 genes-14-01066-t001:** Main biological characteristics of patient one and family.

	Patient P	Son S1	Son S2	Daughter D1	Reference Range
Age (years)	65	31	20	20	n/a
Sex	Male	Male	Male	Female	n/a
Hemoglobin (g/dL)	19.1	14.8	16.9	15.8	11.5–16.0
Hematocrit (%)	56	43	50	47	37.0–47.0
Serum EPO (UI/L)	6.7	Not tested	Not tested	Not tested	2.6–9

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
