# Peer review of "Germline JAK2 E846D Substitution as the Cause of Erythrocytosis?"

_genes, 2023, doi:10.3390/genes14051066_

Round 1

Reviewer 1 Report

The authors aimed to clarify the role of JAK2 E846D in the development of erythrocytosis.

The paper lacks any functional studies and, in several places, confuses or ambiguously defines the boundary between congenital erythrocytosis and acquired primary erythrocytosis – polycythemia vera, a clonal myeloproliferative disease caused by driver oncogenic JAK2 mutations.

Major comments:

1.       The authors do not distinguish between the effect of JAK2 E846D on JAK2 signaling and on the clinical manifestation of erythrocytosis, which is not the same (Blood. 2016; 128(10):1418-23). They insufficiently distinguish between the potential role of JAK2 E846D in the development of relatively mild, benign erythrocytosis vs. malignant polycythemia.

2.       The paper is built upon two reports describing the presence of JAK2 E846D in congenital erythrocytosis cases: Blood. 2016; 128(10):1418-23 and BJH. 2022; 198(5):923-926, however the interpretation of the data from these two publications is inaccurate and, in some aspects, misleading or even incorrect.

E.g., page 2, lines 53: “Furthermore, in a previous study, this same variant had been reported to stimulate the JAK-STAT pathway, but only in association with another variation (2).” and page 9 lines 163-165: “The variant was firstly described in a study as a contributing factor leading to the activation of the JAK/STAT pathway, not by its sole implication but in association with other variations (2).”

This is an incorrect interpretation of the data from the Blood paper were the authors showed (using extensive in vitro and in vivo experiments) that JAK2 E846D alone weakly hyperactivates JAK2/STAT5 signaling, but the cooperation with another weakly activating JAK2 variant (R1063H) is needed for the disease phenotype to develop.

Also, in the BJH paper the authors propose, that other factors besides JAK2 E846D, may play a role in the disease development, because the degree of erythrocytosis varied among affected family members. It is too strong to state that the study from BJH “assigns more damaging role to this variant” (page 6, line 166)

3.    The conclusions of the paper (the last paragraph, page 6, lines 183-185) is not novel. It was already published that the impact of JAK2 E846D is much weaker when compared to JAK2 V617F; and it is known that polycythemia vera is not attributed to heterozygous germline JAK2 variants. However, it has been already shown that the weakly activating germline JAK2 R1063H mutation co-occurs with JAK2 V617F in myeloproliferative neoplasms and amplifies JAK2 V617F signaling (Blood. 2018;132(25):2695-2699).

4.    The possible involvement of JAK2 E846D in congenital erythrocytosis (two positive families detected by the authors) is not sufficiently addressed or discussed. A more extensive genetic study (whole exome sequencing) in patients positive for E846D might help to find other relevant genetic abnormalities that could be functionally assessed including their possible co-operation with JAK2 E846D.

5.    Hypersensitivity of primary cells to EPO and assessment of phospho-STAT5 should be carried out on primary cells in the analyzed family member of patient 2+the patient. Similar analyses should be performed using several unrelated JAK2 E846D-positive and JAK2 E846D-negative healthy subjects identified in the study group. 

6.    Did the authors confirm the germline origin of JAK2 E846D in all positive 760 individuals from the UK biobank? Which method and what material was used to detect the presence of this mutation in this cohort and how the germline origin was confirmed?

7.    The authors also state (page 2, lines 69, 70): “… was used to analyze the hematological parameters (red blood cells count, hemoglobin, reticulocyte counts) of the carriers of JAK2 genetic variants”.  Have the authors evaluated also other (germline) JAK2 variants and what are the results? These data are not presented.

8.    Page 6 (lines 168-174): the authors are discussing normal EPO levels in their erythrocytic patients with JAK2 E846D, but these data are incomplete. For patient 1 no values are provided and the reference range is also missing. In addition, they again compare EPO values to PV (although they patients do not have MPN): “In our study, the two patients with the JAK2 E846D variant had normal serum EPO values, which does not argue in favor of PV: indeed, EPO levels are classically decreased.” Moreover, according to the current WHO criteria for the diagnosis of PV, a subnormal EPO level is only a minor criterion and some PV patients have normal EPO. In addition, their patient 1 is a heavy smoker (one pack/day for 40 years), which may influence his EPO levels. Overall, the EPO data presented are insufficient to conclude that the JAK2 E846D variant plays no functional role.

Minor comments:

1.       Page 1, Line 41: “This condition” refers to PV or congenital erythrocytosis?

2.       Method for the measurements of RCM should be presented.

3.       Figure 1 and Figure 2 should be combined to one figure and the same data should be shown for both families: pedigree, seq. analysis.

4.       The hematocrit and hemoglobin values in Table 1 should be presented as mean ± SD of at least 3 independent measurements over a longer period of time. Reference values should be added as well.

5.       The sentences in lines 111-116 (page 3) are likely misplaced in the text. If they describe genetic analyses in the family of patient 2, they should follow after the presentation of the patient (page 4, lines 121-139).

6.       Line 154 (page 5): the term “classical” JAK2 V617F is not commonly used; should be driver or oncogenic.

Reviewer 2 Report

The authors present an analysis of a pedigree with a reported JAK2 germline variant that was previously proposed to be associated with congenital erythrocytosis.  They clearly demonstrate the lack of association of the variant with the clinical phenotype of erythrocytosis in that single pedigree.  Additionally, they review the impact of the variant when observed in a large UK series and find no confident association with erythrocytosis. 

The manuscript is very well written.  It is succinct, easy to follow and the figures add nicely to the story.  The conclusions are sound. I have no suggested revisions. While the impact of this work is not huge, I found it interesting and think it would be of value to readers of the journal.  

Author Response

We thank the reviewer for his remarks.  

Reviewer 3 Report

Maaziz and colleagues provide an interesting contribution to an expanding field of research in hematology. The study design is appropriate and the results are clearly presented. In my opinion, a more extensive discussion is highly needed; there is evidence (10.1186/s40164-022-00301-1, 10.1002/ajh.26920, 10.1038/s41375-020-0847-4) widely discussing the occurrence of JAK2 unmated erythrocytosis that should be adequately included and commented by the authors. The presented study definitely adds new insights to this theme. 
